# Debris Flow Susceptibility in the Jinsha River Basin, China: A Bayesian Assessment Framework Based on Geomorphodynamic Parameters

Zhenkui Gu<sup>1, 2</sup>, Xin Yao<sup>1, 2</sup>, Xuchao Zhu<sup>3</sup>

<sup>1</sup>Institute of Geomechanics, Chinese Academy of Geological Sciences, Beijing 100081, China

<sup>2</sup>Key Laboratory of Active Tectonics and Geological Safety, Ministry of Natural Resources, Beijing 100081, China

<sup>3</sup>State Key Laboratory of Soil and Sustainable Agriculture, Institute of Soil Science, Chinese Academy of Sciences, Nanjing 210008, China

Correspondence to: Zhenkui Gu (guzhenkui 15@mails.ucas.ac.cn)

Abstract. A major challenge in assessing debris-flow susceptibility at the scale of large mountainous river basins lies in the excessive reliance on simplified topographic metrics. Existing approaches often fail to account for the cascading and dynamically coupled interactions among channel gradient, discharge, and sediment supply. This oversight limits the accuracy and robustness of spatial predictions. To address this gap, we present a novel framework for debris-flow susceptibility assessment grounded in a process-based indicator system derived from geomorphic dynamics, using the Jinsha River Basin as a case study. Our method integrates key parameters that characterize landscape evolution—including stream power, extreme rainfall events, surface erodibility, and sediment connectivity—into a Naïve Bayes probabilistic classification model. By employing kernel functions, we accommodate both continuous and discrete variables, enabling the probabilistic estimation of debris-flow occurrence across small, medium, and large magnitude classes. Model validation across the Jinsha River Basin yields a prediction accuracy of 63%. Notably, empirical testing against the "8·21" Jinyang debris-flow event in 2023 reveals a high degree of spatial agreement between predicted high-risk zones and observed disaster footprints. Feature importance analysis indicates that surface erodibility is the dominant contributor to susceptibility, followed by connectivity, stream power, and extreme precipitation. Approximately 32,000 high-risk gullies (>200 m in length) exhibit a power-law distribution, clustering within a 30 km buffer on both sides of the main stem of the Jinsha and Yalong Rivers in their middle and lower reaches. These regions are shown to be strongly associated with infrequent but high-probability events, which tend to drive large-scale debris-flow disasters. Amid intensifying climate change and the rapid expansion of infrastructure in alpine canyon regions, the dynamic datasets we construct—such as stream power and sediment connectivity—offer a quantitative basis for risk-informed planning and mitigation. This modeling approach represents a scalable and physically grounded paradigm for debris-flow hazard assessment, offering broad applicability to other high-relief mountainous environments worldwide.

Keywords: Debris Flow; Bayesian; Geomorphodynamic Parameters; Jinsha River

#### 1 Instroduction




Debris flows in mountainous regions are characterized by active runoff erosion, significant topographic relief, and the interplay of tectonic uplift and river incision (Qiu et al., 2021; Ciccarese et al., 2020; Ye et al., 2023). These flows are triggered by various processes, including shallow landslides, runoff infiltration, channel mobilization, dam failure, and rapid snowmelt (Qiu et al., 2021; Ciccarese et al., 2020; Ye et al., 2023). Due to their high kinetic energy, debris flows pose significant risks to infrastructure, including roads, bridges, and buildings. Globally, debris flow-prone areas are concentrated in the Pacific Rim fold belt, the Alpine-Himalayan fold belt, and mountainous regions in Eurasia (Ye et al., 2023). In China, regions such as the Gongga Mountains, the western Loess Plateau, and the Jinsha River Basin are particularly vulnerable, with the latter contributing significantly to debris flow disasters in the country (Hu et al., 2020). In recent decades, the ongoing global and regional climate warming has exacerbated the risks associated with debris flows by increasing the frequency of extreme weather events (Lu et al., 2021; Zhao et al., 2021). This highlights the urgent need for research on debris flows. Such research has informed the development of quantitative models, including the power-law relationship between mean intensity and rainfall duration (Coe et al., 2008; Badoux et al., 2009; Oorthuis et al., 2021; Hürlimann et al., 2019; Nikolopoulos et al., 2014), and the linear relationship between surge front velocities and flow depth (Mccoy et al., 2011). These advancements aim to enhance the working principles of monitoring and early warning stations to achieve more accurate debris flow forecasting. However, despite these efforts, debris flow disasters continue to occur, and the role of monitoring stations in regional safety remains limited. From 1999 to 2019, debris flows in China resulted in 4,742 deaths, with an average of 226 deaths per year. In 2010 alone, 2,073 people lost their lives. Notably, on August 7, 2010, a large debris flow in Zhouqu, Gansu, destroyed over 390 buildings (Zhang et al., 2018b); on June 28, 2012, debris flows occurred in ten gullies upstream of the Baihetan hydropower station, including the Aizi gully, resulting in the death or disappearance of 41 people (Hu et al., 2017); and on August 17, 2020, a debris flow occurred in Dayi, Sichuan, blocking a river and causing flooding (An et al., 2022). The uncertainty in the spatiotemporal distribution of extreme precipitation events, combined with insufficient understanding of regional debris flow risk assessment and patterns, has led to the absence of monitoring stations or improper site selection in some high-risk areas, thus limiting the effectiveness of early warning systems (Li et al., 2024).

Currently, numerical simulations based on momentum conservation, mass conservation, and rheological equations are commonly used to model the kinematic characteristics of material flow in debris-flow-prone gullies. These simulations provide key parameters, such as flow velocity and transport volume. However, accurately identifying potential debris flow locations in large-scale areas remains a significant challenge. In the past, there has been a heavy reliance on the direct interpretation of remote sensing images (Yi and Qu, 2018; Lyu et al., 2022; Hu et al., 2017), using abnormal reflectance from surface damage areas after debris flows, such as vegetation, bare soil, and gravel, as indicators. More recently, the accuracy and efficiency of debris flow susceptibility assessments have significantly improved with the development and application of machine learning models. These models are trained using quantitative surface characteristics of debris flow-

prone areas, including slope, aspect, topographic relief and roughness, lithology, and NDVI (Li et al., 2024). However, the development of debris-flow channels is an intense "source-sink" process, where debris and surface water flow along specific slopes, with the valley bottom being the most destructive area. The current assessment systems based on indicators such as slope, lithology, and NDVI mainly reflect the characteristics of the valley slopes on both sides, which do not fully correspond to the dynamic nature of the valley bottom. Therefore, it is necessary to reconstruct the indicator framework and establish a parameter system that better aligns with the physical characteristics of the valley bottom to improve the accuracy of debris flow susceptibility assessments. To address these challenges, we propose an assessment framework based on a Naïve Bayesian model to improve the identification of debris flow locations, timing, and likelihood. Focusing on the Jinsha River Basin, this framework incorporates parameters such as stream power, surface erosion susceptibility, sediment transport connectivity, and the frequency and intensity of extreme precipitation events. The dataset generated by this approach describes the dynamic quantitative characteristics of debris flow gullies and the probability of occurrence, helping us identify many potential debris flow locations previously overlooked. This framework provides practical reference points for site selection in major infrastructure projects and disaster prevention engineering.

#### 2 Study Area


The Jinsha River, a crucial tributary of the Yangtze River, originates in the Tanggula Mountains of China. It traverses several distinct natural regions, including the eastern Qinghai-Tibet Plateau, the northwestern Yunnan-Guizhou Plateau, and the southwestern Sichuan Basin (Fig. 1). The mainstream flows for approximately 2,316 km, with an average gradient of 2.16 ‰, and an annual average discharge of 4,750 m<sup>3</sup>/s, draining a catchment area of about 5×10<sup>5</sup> km<sup>2</sup> (Li et al., 2018). In its upper reaches, the terrain is relatively flat, underlain by continental crust that was formed and recycled during the Paleozoic era. The landscape is characterized by desert meadows, and the valley is wide and shallow, resulting in slow river flow. As the river progresses into the middle reaches, it enters the Indosinian fold belt, where the continental crust formed during the Meso-Cenozoic era. The lower basement consists of ancient Precambrian continental crust (Ma, 2002). The entire river basin lies within a seismically active zone due to ongoing neotectonic activity, characterized by numerous faults and generally fractured rock masses. Precipitation in the region is concentrated between May and October, driven by both southwest and southeast monsoons, with extreme rainfall typically occurring from June to August. The annual average precipitation is 632 mm, increasing gradually from northwest to southeast. However, in areas above 4,000 m, the average annual rainfall drops to just 344 mm, making it the driest region in the Yangtze River basin (Cao et al., 2011). Over the past six decades, river discharge has increased, driven by global warming and the accelerated melting of ice and snow (Liu et al., 2016). The rapid tectonic uplift and river erosion in the region have shaped deep canyon-type landforms, with valley depths exceeding 1,000 m. The dynamic interaction between internal tectonic forces and external erosional processes has contributed to the development of a highly active river system, prone to frequent landslides and debris flows (Liu et al., 2018).

Figure 1: The Jinsha River Basin and Its Adjacent River Systems. Note: The projected coordinate system utilizes the UTM zone 47N (6-degree longitudinal division) based on WGS 84 datum.

# 95 3 Methodology


# 3.1 Data and Preprocessing

This study utilizes a comprehensive set of data sourced from various repositories, including debris-flow surveys, stream discharge records, precipitation data, topographic information, and soil characteristics. The key datasets and preprocessing steps are outlined below. 1)Stream Discharge, Discharge data from hydrological stations are crucial for estimating stream power; 2) Rainfall, the Standardized Precipitation Index (SPI) is computed at daily, monthly, and annual scales using high-resolution, long-term daily grid precipitation data from the ECMWF ERA5-Land product (Period: Jan 1950–present) with a

~9 spatial resolution, which is derived from radar km and satellite-based weather observations (https://cds.climate.copernicus.eu), this data demonstrates superior quality compared to satellite-based precipitation retrieval in similar data products (Xu et al., 2022); 3) Topography, elevation and catchment areas along the longitudinal profile are extracted from SRTM 1" DSM data, which offers a spatial resolution of approximately 30 meters; Critically, the dataset deliberately retains vegetation elevation values, thereby facilitating the acquisition of surface connectivity characteristics under vegetative interference and enabling systematic evaluation of vegetation's modulating effects on debris flow susceptibility mechanisms; 4) Debris Flow Incident Sites, the distribution of debris flow events in China is mapped at a scale of 1:5,000,000 (or Obtain the vector data from Resource and Environmental Science Data Platform, Chinese Academy of Sciences; https://www.resdc.cn/). This map, based on field investigations and depositional markers, provides locations and magnitudes of historical debris flows (Yi and Qu, 2018). Due to spatial discrepancies between some annotated hazard points and the corresponding gully centerlines, it was necessary to manually adjust the positions of these points within the ArcGIS platform prior to the calculation of geomorphic dynamic parameters. This step ensured the alignment of points with relevant geomorphic features and allowed us to retain enough samples for both model training and validation. 5) Soil Characteristics, the China Soil Map-based Harmonized World Soil Database (HWSD v1.2) is used to estimate soil erodibility (K), with a spatial resolution of 250 meters (Wieder et al., 2014). To reduce potential errors, data from flat surfaces are excluded. When implementing the D8 algorithm for depression filling in DEM processing, the filled regions (typically reservoirs or natural depressions) exhibit zero gradient in elevation. Fluid flow through such hydraulically flattened areas undergoes complete kinetic energy dissipation, rendering them non-informative for hydrodynamic investigations. Debris flows are categorized into three classes based on the volume of the accumulation body: small ( $< 1 \times$  $10^4$  m<sup>3</sup>), medium (1 ×  $10^4$ –1 ×  $10^5$  m<sup>3</sup>), and large (> 1 ×  $10^5$ –1 ×  $10^6$  m<sup>3</sup>). Volume estimates account for factors such as debris-flow bulk density, solid particle bulk density, debris-flow duration, and peak discharge (Yu and Tang, 2016). The surface regolith data at a reference depth of 1 meter provide detailed information on the percentage contents of gravel, sand, clay, and organic matter, along with related parameters (Meng and Wang, 2018). This methodological framework ensures an accurate assessment of debris-flow susceptibility by integrating critical environmental and geological factors.

#### 3.2 Modeling Approach







Debris flows are influenced by surface erosion and sediment supply, requiring a thorough consideration and quantification of related factors. Before designing the assessment framework, we identified key indicators with significant physical relevance to Earth's surface processes and made necessary adjustments to produce a three-dimensional visual representation of the numerical values. During the research, we used parameter sequences from debris-flow survey sites as training and testing samples. These parameters include dynamic characteristics of surface rock erosion, sediment connectivity, stream power, and the frequency and severity of extreme precipitation events in highly sensitive debris-flow valleys within the Jinsha River

basin. A Naïve Bayes model was then applied to assess debris-flow probability across daily, monthly, and annual timescales (Fig. 2).

This model calculates the posterior probability of each feature using Bayesian inference based on its prior probability, assigning it to the category with the highest posterior probability. Specifically, if there are m classes (e.g., non-occurring, small, medium, and large debris flows) denoted as  $C_1$ ,  $C_2$ , ...,  $C_k$ , and spatiotemporal variables denoted as  $x_1$ ,  $x_2$ , ...,  $x_5$  (e.g., stream power, erodibility, connectivity, and the severity and frequency of extreme precipitation). According to Bayes' theorem, the posterior probability of a class  $C_k$  given a set of features  $x = (x_1, x_2, ..., x_5)$  can be expressed as:

$$P(C_k|x_1, x_2,..., x_5) = \frac{P(C_k) \cdot P(x_1, x_2,..., x_5|C_k)}{P(x_1, x_2,..., x_5)}$$
 (1)



Here,  $P(C_k)$  denotes the prior probability of class  $C_k$ , which reflects the proportion of samples belonging to that class in the training data. The term  $P(x_1, x_2, ..., x_5 | C_k)$  represents the likelihood, i.e., the joint probability of observing the feature set xxx given that the sample belongs to class  $C_k$ . The denominator  $P(x_1, x_2, ..., x_5)$  is the marginal probability of the features, which remains constant across classes and thus can be omitted when performing class comparisons. In the Naïve Bayes framework, a core assumption is that features are conditionally independent given the class. This allows the likelihood term to be factorized into a product of individual conditional probabilities:

$$P(x_1, x_2, ..., x_5 | C_k) = P(x_1 | C_k) \cdot P(x_2 | C_k), ..., P(x_5 | C_k)$$
(2)

Substituting Equation (2) into Equation (1), the posterior probability simplifies to:

$$P(C_k|x_1, x_2, ..., x_5) \propto P(C_k) \cdot \prod_{i=1}^5 P(x_i|C_k)$$
 (3)

For classification, a new sample  $x = (x_1, x_2, ..., x_5)$  is assigned to the class with the highest posterior probability:

$$\widehat{y} = \arg \max_{C_k} \left[ P\left( C_k \cdot \prod_{i=1}^5 P(x_i | C_k) \right) \right] \tag{4}$$

The likelihood estimation methods are based on the normal distribution function (for continuous variables), frequency statistics (for discrete variables), and the Bernoulli equation (for binary outcomes, i.e., 0 or 1). In this study, the conditional probability density  $P(x_i|C_k)$  is estimated using a kernel density estimation (KDE) approach defined as:

155 
$$P(x_i|C_k) = \frac{1}{n_k h_i} \sum_{j=1}^{n_k} K(\frac{x_i - x_{i,j}^k}{h_i})$$
 (5)

Where  $n_k$  is the number of training samples in class  $C_k$ ,  $x_{i,j}^k$  is the  $j^{th}$  observation of feature i in class  $C_k$ ,  $h_i$  is the bandwidth for feature i, and K denotes the kernel function—typically the Gaussian function. It is important to note that when feature values are approximately continuous across their ranges, the Gaussian kernel can be directly applied to estimate the probability density. However, in cases where certain features are relatively discrete, a preprocessing step is introduced to estimate a smoothed continuous distribution using the kernel function, thereby enabling the application of Equation (5) under the continuous assumption. The mutual information measure permits analysis with both continuous and categorical variables and has been widely adopted in the literature; we therefore select this metric (Blanquero et al., 2021; Kinney et al., 2010). It quantifies the information about variable X contained in variable Y, defined formally as:

$$I(X,Y) = \iint P(x,y) \log(\frac{P(x,y)}{P(x)P(y)}) dxdy$$
(6)

The attribute value for a point within the basin is computed as the average value of the upstream confluence interval, using the following formula:

$$\overline{F}_{j} = \frac{\sum_{i=1}^{n} E_{i,j}}{N_{i,i}} \tag{7}$$

In the formula,  $E_{i,j}$  is the attribute value of the j<sup>th</sup> parameter at point i, and N is the number of corresponding grids. Due to the algorithm's resilience, this model is not susceptible to missing data, and the discriminant effect is steady (Mu et al., 2021; Soomro et al., 2022). Following this scheme, the probability of debris flow at various sizes and durations can be determined to produce a more realistic and understandable illustration of debris-flow susceptibility.

Figure 2: Study Implementation Framework.

## 3.3 Quantitative parameters

# 3.3.1 Stream power and its gradient

Stream power (W/m) is the rate at which runoff's gravitational potential energy is transformed into kinetic energy (Pérez-Peña et al., 2009). Its ratio (ω, W/m²) to river width may be used to quantify runoff erosivity to river channels (Bagnold, 1960). When stream power increases throughout the channel, the value is higher than 0, and runoff erodes; when stream power is reduced downstream, the value is less than 0, indicating an energy-dissipating stretch and sediment deposition occur. Erosion and deposition are balanced at 0 (Lea and Legleiter, 2016). In fluvial systems across low-relief terrain, stream power per unit channel length (Ω, in W/m) is dominantly governed by discharge (Q, m³/s) and channel width, as longitudinal gradients (S, %) exhibit minimal spatial variability. We quantify the spatial distribution of erosional potential within the valley using specific stream power (ω, W/m²), calculated as:

$$\omega = \Omega/L \tag{8}$$

where L represents the reach length (m). Given that  $\Omega$  is functionally linked to discharge and gradient ( $\Omega = \gamma QS$ ), and discharge can be parametrized as a power-law function of contributing catchment area ( $Q = aA^b$ ), Equation 3 is expanded into:

$$\omega = \frac{\gamma a A^b S}{L} \tag{9}$$

Here,  $\gamma$  denotes the specific weight of the fluid (N/m³). While clean water flows adopt  $\gamma = 9800$  N/m³, debris-laden flows require an amplified value (Reference value:  $\gamma = 16,000$  N/m³) to account for elevated bulk densities (1000–2400 kg/m³) and enhanced erosive loads. Coefficients a and b were calibrated via nonlinear regression against gauge-derived Q and A

Figure 3: Diagram of numbering reaches. Note: The reach between the two gully junctions is considered a gradient cell.

Figure: 4 The calculation process of stream power and its gradient.

measurements. Crucially, this formulation focuses on isolating baseline erosional drivers, deliberately excluding transient sediment feedback to align with the study's scope of identifying first-order geomorphic controls.

Stream power computation involves depression-filling, slope aspect, catchment area, channel numbering, and elevation extraction. Using a threshold of 1 km<sup>2</sup>, we extracted the longitudinal profile of the valley and its catchment areas (m<sup>2</sup>), and the channel gradients were computed using the first order fitting function (Fig. 3):

$$\begin{cases}
 h_{1,i} = k_1 L_{1,i} + c_{1,i}, & R_1^2 \\
 h_{2,j} = k_2 L_{2,j} + c_{2,j}, & R_2^2 \\
 & \dots \\
 h_{n,m} = k_n L_{n,m} + c_{n,m}, & R_n^2
 \end{cases}
 \tag{10}$$

where n is the number of reaches;  $h_{n,m}$  denotes the elevation of point m; C is the corresponding constant;  $k_n$  is the ratio of the calculation;  $R_n^2$  is the linear fitting coefficient of the reach;  $L_{n,m}$  is the length (m) between the  $m_{th}$  point of a certain reach and its gully head; and i, j,..., m has the same meaning as m but represents river segments with different lengths, that is, the distance between the intersection of two confluence points (Fig. 3). The calculation process is shown in Fig. 4.

## 200 3.3.2 Index of connectivity (IC)

Connectivity reflects the topographic resistance of detrital material on a mountain as it is transported. The transport mechanism of detrital materials will change due to the tight relationship between the upslope component  $(D_{up})$  and the downslope component  $(D_{dn})$  with topographic variations. The following equation is as follows:

$$IC = Log_{10}(\frac{D_{up}}{D_{du}}) \tag{11}$$

where IC is defined in the range of  $[-\infty, +\infty]$ , with greater IC values indicating higher connectivity. The upslope component  $(D_{up})$  describes the potential for the downward routing of sediment produced upslope and is estimated as follows:

$$D_{\rm up} = \overline{W} \overline{S} \sqrt{A} \tag{12}$$

where  $\overline{W}$  is the average weighting factor for the upslope contributing area,  $\overline{S}$  is the mean slope (%), and A is the size (m<sup>2</sup>). The downslope component ( $D_{dn}$ ) considers particles' flow path lengths to reach the nearest target or sink. It is expressed as follows:

$$D_{\rm dn} = \sum_{i} \frac{d_i}{W_i S_i} \tag{13}$$

where  $d_i$  is the length of the flow path along the ith cell according to the steepest downslope direction (m), and  $W_i$  and  $S_i$  are the weighting factor and slope of the  $i^{th}$  cell, respectively (Jing et al., 2022). Determining weighting factors within a watershed uses the standardized roughness index (SRI) or land use classification data (Zanandrea et al., 2020). The determination of weights in this paper is based on the standardized roughness index (RI), which is calculated as the standard deviation of the difference between the nonsmoothed and smoothed DTM and can represent vegetated regions(Zanandrea et al., 2020). The RI values provide valuable surface roughness information computed in an n×n cell moving window over the residual topography grid. The RI is defined as follows:

$$RI = \sqrt{\frac{\sum_{i=1}^{25} (x_i - x_m)^2}{25}}$$
 (14)

where  $x_i$  is the value of each cell of the residual topography within the moving window, and  $x_m$  is the mean of the n×n cell values. Here, we used nine as the number of processing cells within the 3×3 cell moving window. The W value is typically calculated from the RI according to the methodology defined by the following:

$$W_{\rm RI} = 1 - \left(\frac{RI}{RI_{\rm Max}}\right) \tag{15}$$

where RI<sub>Max</sub> is the maximum RI value in the study area.

## 225 3.3.3 Extreme precipitation identification

Extreme precipitation (or wetting) events are identified using the run theory (Huang et al., 2021; Yevjevich, 1969). We used McKee's standardized precipitation index (SPI) from 1993 to characterize the precipitation probabilities and observed extreme precipitation events at three scales: daily, monthly, and annual (Table 1). In the SPI, a two-parameter gamma probability density function is used to explain the frequency distribution of precipitation:

$$\mathbf{g}(x) = \frac{1}{\beta^{\alpha} \Gamma(\alpha)} x^{\alpha - 1} e^{\frac{x}{\beta}} \tag{16}$$

where x is the precipitation accumulation, and  $\Gamma(\alpha)$  is the gamma function. The gamma distribution's shape and scale parameters,  $\alpha$  and  $\beta$ , may be calculated using the most excellent likelihood method (Edwards, 1997). Under certain conditions, the cumulative probability G(x) can be reduced to the so-called incomplete cumulative gamma distribution function,  $t = \frac{x}{\beta}$ :

$$G(x) = \frac{1}{\Gamma(\alpha)} \int_0^x t^{\alpha - 1} e^{-t} dt$$
 (17)

Since Eq. (12) is invalid for zero precipitation (x=0), the cumulative probability distribution, including zeros, may be stated as H(x)=q+(1-q) G(x), where q and 1-q are the probabilities of zero (x=0) and nonzero (x≠0) precipitations, respectively. The SPI is computed by changing H(x) to a zero-mean, one-variance normal distribution. Positive SPI levels imply moist periods, whereas negative values suggest dry periods (Farahmand and Aghakouchak, 2015). The severity of precipitation can be described as the total of the SPI across the length of numerous single severe rainfalls.

$$S_{ij} = \left(\frac{1}{m} \sum_{i=1}^{m} |SPI_i|\right)_i \tag{18}$$

where m is the number of extreme wetting events, indicating wetting occurrences dominated by precipitation.

Table 1 Category of standardized precipitation index (SPI) based on range values (Dutta et al., 2015; Mckee et al., 1993)

| SPI Range     | Category       |  |
|---------------|----------------|--|
| +2 to more    | Extremely wet  |  |
| 1.5 to 1.99   | Very wet       |  |
| 1.0 to 1.49   | Moderately wet |  |
| -0.99 to 0.99 | Near Normal    |  |
| -1.0 to -1.49 | Moderately dry |  |
| -1.5 to -1.99 | Severely dry   |  |
| -2 to less    | Extremely dry  |  |

#### 3.3.4 Erodibility (K)

Erodibility (*K*) is a surface erosion factor related to the concentration of organic materials, sand, mud, and gravel in weathered accumulations. A higher number suggests a more easily degraded surface nature. It is commonly represented as the number of soil particles lost due to precipitation erosivity per unit of time in a standard area. The models used to calculate K include Nomograph (Wischmeier et al., 1971), EPIC(Sharpley and Williams, 1990), Torri (Torri et al., 1997), Shirazi(Shirazi et al., 1988), and Wang (Wang et al., 2013). As it is more widespread in hilly places, the EPIC model (Erosion/Productivity Impact Calculator) was utilized to estimate erosion in this study. The model can be expressed as follows:

$$K_{\text{EPIC}} = \left[0.2 + 0.3e^{-0.0256\varphi_{\text{sa}}\left(1 - \frac{\varphi_{\text{si}}}{100}\right)}\right] \times \left(\frac{\varphi_{\text{si}}}{\varphi_{\text{cl}} + \varphi_{\text{si}}}\right)^{0.3} \times \left(1 - \frac{0.25\varphi_{\text{oc}}}{\varphi_{\text{oc}} + \varphi^{3.72 - 2.95\varphi_{\text{oc}}}}\right) \times \left[1 - \frac{0.7(\varphi_{\text{cl}} + \varphi_{\text{si}})}{\varphi_{\text{cl}} + \varphi_{\text{si}} + e^{-5.51 + 22.9(\varphi_{\text{cl}} + \varphi_{\text{si}})}}\right]$$
(19)

where  $\varphi_{sa}$ ,  $\varphi_{si}$ ,  $\varphi_{oc}$  and  $\varphi_{cl}$ (%) are the sand, silt, organic carbon and clay contents, respectively (Sharpley and Williams, 1990).

#### 4 Results





## 4.1 Mapping of High-Energy Valleys and Erosion Dynamics

Stream power is a critical parameter in erosion processes, as it reflects the rate at which gravitational potential energy is converted into kinetic energy, closely linking it to the channel gradient. In the Jinsha River basin, most areas have a channel slope of less than 5.63%, with regions of steeper gradients predominantly concentrated in the middle and lower reaches of the Jinsha and Yalong Rivers, within approximately 30 kilometers of the riverbanks (Fig. 5a and 6). According to geomorphological evolution principles, in the initial stages of erosion, the longitudinal profile of the valley typically follows a straight-line form. As the erosion process progresses, this profile gradually becomes more curved, and eventually, the mountains are reduced to a peneplain. Throughout the different stages of this process, the valley's longitudinal profile can be best represented by four functions: linear, exponential, logarithmic, and power, in the following sequence: linear → exponential → logarithmic → power (Ohmori and Saito, 1993; Ohmori, 1991; Rãdoane et al., 2003). In contrast, the longitudinal profiles of most valleys in the basin display distinct linear characteristics, with an average linear fitting coefficient (R<sup>2</sup>) exceeding 0.94 (Fig. 5b). This suggests that most valleys in the basin are still in the early stages of erosional evolution. To quantify stream power, we estimated the flow parameters and gradients at each grid location, converting them into stream power values (Fig. 5d). In Fig. 5c, we categorized river segments by different stream power intervals. Figures 5e, h, and k show the geographical locations of erosion and deposition along the downstream river sections. Our analysis revealed that effective erosion in the Jinsha River basin is primarily concentrated in the middle and lower reaches, with tributaries on both sides exhibiting stronger erosional activity (Fig. 5a). By using an average stream power gradient threshold of 1×10<sup>-4</sup> W/m<sup>2</sup>, we identified high-energy valleys and validated this threshold using debris flow fans as geomorphic markers (Fig. 5f-1 and 5f-2). We then quantified the number of high-energy valleys at various buffer distances along the Jinsha and Yalong Rivers, which revealed a significant power-function relationship (Fig. 6). The total number of valleys longer than 200 meters is approximately 32,000 (Valley segments shorter than 200 m and disconnected gullies were excluded from statistical aggregation due to resolution limitations.). These valleys, requiring substantial driving forces for debris flows, are likely to pose significant disaster risks.

Figure 5: Spatial Distribution Characteristics of Runoff Erosion Activity: (a) Channel gradient; (b) Linear fit coefficients of the longitudinal profile; (c) Length of river segments across different stream power intervals; (d) Stream power distribution; (e) Stream power gradient; (f) External dynamics in a typical debris flow Channel (f-1) and its geomorphic landscape (f-2); (g) Number of erosion and deposition grids in the mainstream of the Jinsha River; (h) Longitudinal profile of the Jinsha River and stream power gradient along the river; (k) A typical high-energy watershed environment. Note: In this study, we calculated the gradient values, stream power, and power gradients for all river reaches. Due to the extensive spatial data involved, we applied interpolation techniques to simplify the results for easier interpretation by readers, as shown in (a), (b), (d), and (e). Image credit: Z. Gu.

Figure 6: Characteristic statistics of High-Energy valley: (a)Variation in the Number of High-Energy Reaches with Channel Buffer Distance; (b) Debris flow investigation points in various stream power gradient intervals. Note: High-energy valleys are defined here as those with a stream power gradient greater than  $1.3 \times 10^{-4} \text{W/m}^2$  and the threshold is defined according to the inflection point of the trend change of the fitted curve. This chart displays the count of high-energy valleys within a 200m buffer along the Jinsha River and Yalong River, across a range of buffer widths, specifically including those with a stream power gradient exceeding  $1.3 \times 10^{-4} \text{ W/m}^2$ .

## 4.2 Variations in Surface Erodibility and Connectivity




The formation and transportation of debris flow source material are significantly influenced by surface erodibility and terrain connectivity. During the short period of debris flow formation, an equilibrium is often established between the supply of eroded material to the river and the river's capacity to transport and deposit these materials. The source material typically originates from loose debris triggered by earthquakes, landslides, or shallow landslides, which evolve into unconfined debris (mud) flows. In flatter regions, stable accumulation occurs, disrupting surface connectivity. As shown in Fig. 7, areas with high erodibility in the Jinsha River basin are primarily concentrated in the downstream regions, where the erodibility factor (K) typically exceeds 0.245 t·ha·h·(ha·MJ·mm) <sup>-1</sup>. These regions are characterized by high clay content and low organic matter. The connectivity of these areas follows a distinct pattern, with lower values in the source regions and higher values in the middle and lower reaches. The Index of Connectivity (IC) values range from -2.47 to 1.17, with high-connectivity zones mainly found in the middle sections of the Jinsha River and along both sides of the Yalong River. In these high-connectivity zones, IC values generally exceed 2.68, which corresponds to the spatial distribution of high-energy or high-gradient valleys. The transition in connectivity between valley slopes and valley bottoms shows a clear decline in values, from high at the slopes to low at the valley bottoms. Deeply incised valleys typically exhibit low connectivity, with the valley bottom often having lower connectivity than the adjacent valley branches. These low-connectivity regions are highly prone to sediment accumulation, which can lead to the formation of barrier dams.

Figure 7: Spatial Variation Characteristics of Surface Erodibility and Connectivity: (a) Spatial distribution of erodibility; (b) Composition of materials within different erodibility ranges; (c) Surface connectivity of the Jinsha River basin; (d) Combined profile of connectivity and elevation (red: mean; black: maximum; blue: minimum).

#### 4.3 Variations in Extreme Precipitation Events and Implications for Debris Flow Risk




We identified extreme precipitation events in the Jinsha River basin over the past decade (2010-2020) using the Standardized Precipitation Index (SPI) on daily, monthly, and yearly time scales, as well as the Run Theory. Figure 8 illustrates that the frequency of extreme precipitation events in any given area is generally fewer than 22 occurrences. The middle and lower reaches of the Jinsha River are identified as high-frequency zones for extreme rainfall events. However, as the observation time scale increases, a noticeable shift of these high-frequency areas towards the upstream regions occurs. The observed spatiotemporal decoupling—wherein extreme precipitation hotspots shift across daily, monthly, and annual scales highlights mechanistic divergence between stochastic microscale forcings (e.g., terrain-modulated convection) and deterministic macroscale controls (e.g., orbital cycles), thereby manifesting intrinsic instability in event patterning. This spatial shift suggests that the pattern of extreme precipitation events is not stable over time. Figures 8b, e, and h display the severity of precipitation events under daily, monthly, and yearly observation scales. The severity of these events is negatively correlated with the frequency of extreme precipitation events (Figs. 8c, f, and i). Consequently, in regions with fewer occurrences of extreme precipitation, when debris flows do occur, they may be more destructive in terms of scale and intensity than in areas with higher frequencies of extreme precipitation events. Time-series statistical analysis reveals that 2014 experienced a higher number of extreme precipitation events, with a decrease in frequency observed starting from 2015 (Fig. 9a). This trend indicates a declining risk of debris flows in the Jinsha River basin in recent years. Extreme precipitation events most frequently occur in July, accounting for approximately 30% of the total annual occurrences (Fig. 9b). The severity of major precipitation events shows minimal interannual variation (Fig. 9c).

Figure 8: Frequency, Severity, and Correlation of Extreme Precipitation in the Jinsha River Basin (2010–2020). Note: Panels (a), (d), and (g) illustrate the number of extreme precipitation events from 2010 to 2020 at daily, monthly, and yearly observation scales, respectively; Panels (b), (e), and (h) demonstrate the severity of extreme precipitation events at three observation scales: daily, monthly, and yearly. Panels (c), (f), and (i) depict the severity of extreme precipitation under the corresponding conditions.

Fig. 9 Temporal Characteristics of Extreme Precipitation Frequency and Severity in the Jinsha River Basin: (a) Interannual variation in the frequency of extreme precipitation from 2010 to 2020; (b) Monthly variations in the frequency of extreme precipitation; (c) Severity of extreme precipitation events.

#### 4.4 Probability of Debris Flow Occurrence at Different Observation Scales



The occurrence probabilities of small, medium, and large debris flow events under daily, monthly, and yearly observation scales are presented in Figure 10. The estimation results show that medium- and small-sized debris flows are more prevalent in the basin. During the disaster formation process, the relative importance of various factors contributing to debris flow risk decreases in the following order: surface material erodibility > connectivity > stream power > extreme precipitation frequency and severity (Fig. 11b). To explore the variability in disaster risk, we constructed a Taylor diagram to evaluate the differences in risk across different time scales. This diagram provides a visual comparison of risk deviations at the monthly and yearly scales relative to the daily scale, characterized by standard deviation, root mean square error (RMSE), and correlation coefficient (Fig. 11c). We found that the standard deviation and RMSE for large debris flows at the yearly scale are significantly smaller compared to the other two categories, suggesting that the risk of large debris flows exhibits relatively stable spatial and temporal patterns. Based on these findings, we can conclude that the temporal and spatial stability of debris flow occurrence probabilities in the Jinsha River basin follows this order: large debris flows > small debris flows > medium debris flows.

Figure 10: Probability of Debris Flow Occurrence in the Jinsha River Basin. Note: Panels (a), (b), and (c) represent the probabilities of small debris flows occurring at daily, monthly, and yearly scales, respectively; Panels (d), (e), and (f) depict the probabilities of medium-sized debris flows under the same three time scales; Panels (g), (h), and (i) illustrate the probabilities of large debris flows occurring at daily, monthly, and yearly scales; Panels (k), (l), and (m) show the probabilities of no debris flow

#### 330 4.5 Verification of Disaster Probability Maps with Actual Cases




Model robustness was rigorously validated using an independent dataset preserved from the initial training cohort, with the average accuracy benchmarking at 63% (Fig. 11a). To validate the accuracy of the disaster probability maps, we reviewed news reports of recent debris flow events in the Jinsha River basin and compared them with our evaluation results(Times, 2023a; Times, 2023b). One such event occurred in the early morning of August 21, 2023, when a flash flood-debris flow impacted the Yanjiang Expressway JN1 project section in Lugao Town, Jinyang County, Liangshan Prefecture. The site, managed by Shudao Group, is in the lower reaches of the Jinsha River. According to reports, heavy rainfall persisted for nearly 10 hours prior to the disaster, with accumulated precipitation reaching 160 mm (Fig. 12). Lugao Town (Fig. 13) was the hardest hit, with four confirmed fatalities and 48 missing individuals at the time of reporting. Tragically, in the months

Figure 11: Characteristics of the Probabilistic Model for Debris Flow Occurrence: (a) Confusion matrix; (b) Ranking of covariate importance; (c) Comparisons between daily, monthly, and annual observational scales. Note: SMD, MMD, and GMD represent the deviations of small, medium, and large debris flow occurrence probabilities at the monthly scale relative to the daily scale, respectively; SYD, MYD, and GYD represent the deviations of small, medium, and large debris flow occurrence probabilities at the annual scale relative to the daily scale, respectively.

following the event, all the missing persons were confirmed dead, raising the total death toll to 52. When compared with the probability map we created, the likelihood of a medium-scale debris flow occurring at this location was found to exceed 80%, significantly higher than the surrounding areas (Fig. 13b and f). This supports the accuracy of our model, as it predicted the occurrence of debris flow in a high-risk zone. In the aftermath of the disaster, the local geomorphic landscape was significantly altered (Fig. 13d and e), likely due to a combination of accumulated loose sediment, heavy precipitation, and the presence of a high-energy valley. While the probability of small, large, and super-large-scale debris flows in this area was relatively low (Fig. 13a and c), it is important to note that this does not imply safety in all areas outside the event zone. Our model also identified other high-risk zones in Jinyang County and its surroundings, highlighting the need for enhanced disaster risk preparedness in the future (Fig. 13b).

Figure 12: Precipitation Changes in Jinyang County, Sichuan Province, China, Since 00:00 on August 20, 2003.

Figure 13: Analysis of the "8·21" Debris Flow in Jinyang County Based on Daily Scale Probability of Occurrence: (a) Probability of small debris flow; (b) Probability of medium-sized debris flow; (c) Probability of large debris flow; (d) Photos of the site before the disaster; (e) Photos of the site after the disaster; (f) Location of the disaster on a satellite image. Panels d and e were taken by Z. Gu.

#### 5 Discussion








#### 5.1 Impact of Temporal Observation Scale Changes on the Assessment

Precipitation characteristics are among the most dynamic and least predictable factors influencing debris flow formation. In general, climate change on an annual scale is often determined by ocean-atmosphere coupling oscillations, solar radiation variations, etc; climate change on a monthly scale is mainly determined by the seasonal periodic variations of Earth's orbit; climate change on a daily scale is influenced by local factors such as the modulation of solar radiation forcing by the diurnal cycle, valley wind circulation, land-sea wind oscillation, and human activities. An extreme weather event is often the result of the superposition of the effects of different levels of dynamic factors (Da Silva and Haerter, 2025; Wen et al., 2022; Bi et al., 2023; Ombadi et al., 2023). There are substantial differences in precipitation characteristics across different temporal observation scales. These differences significantly affect our understanding of debris flow susceptibility, suggesting that both the spatial extent and precipitation variables influencing debris flow risk may vary depending on the time scale of observation. We observed an inverse relationship between the frequency and severity of extreme precipitation events, along with notable spatial inconsistencies in the Jinsha River basin at daily, monthly, and annual time scales. Specifically, as the observation scale increased, the number of extreme precipitation events and the extent of high-incidence areas both decreased and shifted (Fig. 9). These findings suggest a pattern in which extreme precipitation events are more frequent, shorter in duration, and more localized on shorter observation scales. In contrast, on longer time scales, these events are less frequent but tend to cover broader spatial and temporal extents. This pattern aligns with broader changes in climate elements such as temperature, wind, and atmospheric pressure (Mckitrick and Christy, 2019), reflecting the complex dynamics of the surface environmental system. Daily precipitation variations are heavily influenced by factors such as diurnal temperature fluctuations, local topography, wind patterns, vegetation cover, and human activities, leading to high variability and low regularity in regional climate change. In contrast, monthly variations are more strongly influenced by seasonal changes driven by Earth's orbital fluctuations, exhibiting clear periodicity and recurrence patterns. For effective disaster preparedness, it is crucial to focus on areas where debris flow susceptibility remains consistent across different time scales. These regions indicate relatively stable spatial and temporal risk, with more predictable probability values. Furthermore, before a debris flow can form, rainfall must undergo processes such as interception, infiltration, and convergence with the vegetation and soil layers to generate sufficient erosive force—processes that inherently require time. Therefore, assessing debris flow susceptibility under different temporal observation scales can help mitigate bias from response time differences, leading to more accurate risk assessments.

# 5.2 Changes in Debris Flow Susceptibility Influenced by Climate Change

The rapid uplift of the Tibetan Plateau within the Jinsha River basin has caused widespread stratigraphic fracturing, destabilizing rock masses and creating favorable conditions for accelerated weathering and gravitational erosion (Zhu et al., 2021; Li et al., 2020). This process has contributed to the accumulation of debris and the formation of highly undulating

terrain, creating a high-energy environment conducive to debris flow development. These geomorphological features also play a key role in controlling the spatial distribution of debris flow-prone areas. Between 2000 and 2015, China experienced 10,927 debris flow disasters, accounting for 36.14% of fatalities from geological hazards (Wei et al., 2021; Zhang et al., 2018a). However, given the vast geographical span of the Jinsha River basin, which covers multiple natural zones with significant spatial and temporal climatic variations, the locations and frequency of such disasters may shift under the influence of global warming (Wei et al., 2021).







The IPCC's 5th Assessment Report indicates a global average surface temperature increase of approximately 0.85°C between 1880 and 2012, with the warming more pronounced in the Northern Hemisphere. The past 30 years have likely experienced the highest temperatures in the last 1,400 years. According to the Clausius-Clapeyron relation, for every 1°C rise in global temperature, the intensity of extreme precipitation increases by 7%, by 15% in high-altitude areas, and precipitation variability rises by 5% (Zhang and Zhou, 2020; Ombadi et al., 2023). We recognize the broad significance of this conclusion, but this does not imply that the basin strictly adhered to this rule during the research period. For at least since 2014, the frequency of extreme precipitation has shown a decreasing trend. This suggests a more uneven temporal distribution of precipitation, with greater fluctuations between wet and dry periods, and an expanded range of precipitation intensities. Two primary theories explain the increase in extreme precipitation events. First, climate warming leads to higher atmospheric moisture content and a slowdown in atmospheric circulation, causing low-pressure systems to remain stationary. Second, weakened summer atmospheric circulation causes it to become slower and more erratic, resulting in prolonged heatwaves and droughts (associated with high-pressure systems) and extended periods of heavy rainfall (associated with low-pressure systems). In China, the intensity and frequency of extreme precipitation events, particularly in the southern regions and the Yangtze River basin, have significantly increased from 1970 to 2018 (Li et al., 2022). These changes have altered the hydrological cycle, leading to shifts in the spatial and temporal distribution of water resources, as well as changes in the overall quantity of available water resources (Wu et al., 2020). As a result, the susceptibility of disaster-prone environments has increased. The Jinsha River basin, in line with general climate trends, has seen increases in temperature, precipitation, and runoff between 1972 and 2017, primarily driven by ice melt and precipitation (Wu et al., 2020). This has caused a significant rise in streamflow from May to June, peaking in July (Fig. 10b). Consequently, this period is critical for debris flow preparedness. Previous studies indicate that precipitation in the Jinsha River basin follows a distinct wet and dry cycle with minimal interannual variability (Song et al., 2012). However, this pattern primarily reflects general precipitation trends, while extreme rainfall events exhibit marked interannual fluctuations (Fig. 9). Future projections for extreme precipitation indices in China suggest a consistent upward trend, with a slight decrease in the number of consecutive dry days (CDD). The growth rate of these indices is expected to accelerate over the coming decades, extending into the middle of this century. Changes in thermal (temperature) and dynamic (circulation) factors are likely contributors to the increased intensity and frequency of future precipitation events (Guo et al., 2018). Recent precipitation simulations for the Yalong River basin under future warming scenarios suggest that the region may experience more frequent and intense precipitation events, which would increase the likelihood of debris flows(Guo et al., 2018). While heavy precipitation typically results in flooding in plains, it can have catastrophic consequences in high mountain valleys, such as those found in the Jinsha River basin(Zhao et al., 2021). Therefore, the susceptibility areas identified in Figure 10 should be prioritized for disaster prevention efforts.

## 5.3 Interaction Between Reservoir Operations and Debris Flow Activity

The Jinsha River basin is rich in hydropower resources, with an exploitable capacity of approximately 1.1 × 10<sup>8</sup> kilowatts, 420 making it one of China's strategically significant hydropower bases. Several large hydropower plants have already been constructed, including Wudongde (dam height: 270 m), Baihetan (289 m), Xiluodu (285.5 m), and Xiangjiaba (88.2 m), with a total installed generation capacity exceeding 4.2 × 10<sup>6</sup> kilowatts. Additionally, numerous smaller hydropower plants are either operational or in the planning stages along the main streams of the Jinsha and Yalong Rivers (Fig. 4). The development of hydropower has significantly altered the river valley landscape, transforming it from one primarily shaped 425 by runoff and erosion into a series of reservoirs extending hundreds of kilometers. Debris flows bring substantial sediment into these reservoir areas, leading to complex interactions between reservoir water levels and debris flow activity. The presence of dams raises the water level, elevating the base level of erosion, which reduces the erosive and incisional forces acting on the valleys along the reservoir areas of the Jinsha River. However, the sediment carried by debris flows contributes to soil and water conservation within the reservoirs (Schmidt et al., 2019), effectively reducing sediment flux and intercepting 430 71.4% of the sediment in the Yangtze River(Lu et al., 2019), surpassing the impacts of land reclamation and landslides caused by agricultural activities. For example, the average annual sediment load at the Panzhihua station increased by 42.4% from 1966–1984 to 1985–2010, primarily due to mineral extraction and deforestation. However, this was followed by a 75.9% decrease from 2011–2015, attributed to the operation of cascade reservoirs in the middle Jinsha River basin since 2010 (Li et al., 2018). Such fluctuations in sediment load can significantly impact the lifespan of the reservoirs. The long-term interplay 435 between regional geology, geomorphology, and hydrology will be shaped by this reciprocal feedback. Notably, nearly all completed and planned reservoirs along the Jinsha and Yalong main streams are situated in areas highly susceptible to debris flows, as identified in this study (Fig. 8a). The number of debris-flow channels exhibits a multiplicative power function relationship with their distance from the mainstream channel, with a distinct trend change occurring within a 5 km radius of the reservoir area (Fig. 6). The relatively dense distribution of debris-flow channels in this zone highlights the significant 440 interaction between reservoir operations and debris flow activity.

#### 5.4 Response to Debris Flow Hazards


The distribution of debris flow gullies in the middle and lower reaches of the Jinsha River is notably dense (Fig. 8a), and the challenges associated with responding to debris flow disasters are exacerbated by global climate change and the development of engineering infrastructure. The occurrence of debris flows has disrupted river ecosystems, making the scientific management of these hazards a pressing societal concern. Effective responses to debris flow disasters must consider the principles of geomorphological evolution and human safety, utilizing the specific spatial and energy characteristics of the

affected areas. Currently, a combination of check dams, ecological engineering, and management practices is widely adopted to mitigate the impacts of debris flows on critical infrastructure and residential areas. These measures include constructing check dams and dredging channels at the mouths of debris flow gullies, creating terraces, afforesting catchment areas, and installing monitoring and early warning systems (Xiong et al., 2016). Globally, building check dams in potential debris flow gullies is recognized as one of the most effective disaster prevention methods (Gao et al., 2022; Chong et al., 2021). This intervention modifies the micro-environment of river valleys in hydrological, geomorphological, and ecological dimensions. In the initial stages, check dams serve multiple functions: storing water, reducing runoff peaks, slowing flow velocity, promoting seepage, and recharging groundwater. Additionally, the dams trap organic matter and sediment, contributing to carbon sequestration and sediment retention. As silt accumulates, the topography upstream of the check dam gradually flattens, creating favorable conditions for vegetation growth and fostering ecological restoration in the local environment (Xiong et al., 2016). Over time, this process can transform debris flow gullies into ecological corridors, directly reflected in reduced surface connectivity and adjustments in river power. A critical challenge in debris flow control is identifying optimal locations and determining the appropriate scale for check dam construction. During dam construction, structures must be designed to accommodate peak flow from potential debris flows. However, for many debris flow gullies, the necessary engineering parameters are often derived from industry-standard formulas, which may be limited by regional variations and insufficient observational data. The findings of this study provide valuable insights into the spatial locations and occurrence probabilities of debris flow-prone valleys in the Jinsha River basin. Beyond merely identifying areas with high debris flow density, this research offers data on stream power, gradient values, surface connectivity, and the probability of debris flow events in specific channels. This enables the precise identification of high-energy valleys and the targeted monitoring and management of these areas. In the context of global climate change, although controlling the frequency and intensity of extreme precipitation events may be challenging, disaster risk areas can be more effectively identified using the debris flow probability maps generated in this study (Fig. 10). High-risk zones of river power and connectivity can be pinpointed from Figures 5d and 7c, allowing for the accurate determination of locations for constructing silt dams. The scale of dam construction can then be optimized based on the relationship between soil erodibility, sediment connectivity, river power, and the observed effects of existing check dams of varying sizes.

## **6 Conclusions**







Accurate delineation of the spatiotemporal window of debris flow occurrence remains a fundamental challenge in mountain hazard assessment and a prerequisite for designing effective disaster mitigation and ecological restoration strategies. Existing large-scale susceptibility evaluations are often constrained by an overreliance on simplistic topographic indicators, limiting their spatial resolution and predictive reliability. Here, we develop a geomorphodynamic Parameters' system that captures the source-to-sink physical processes governing debris flow generation and implement a Naïve Bayesian model to quantify the probabilistic occurrence and spatial heterogeneity of

debris flows across multiple scales in the Jinsha River Basin. The model achieves an average accuracy of 63% and demonstrates high spatial localization precision, as validated by the catastrophic "August 21" 2023 debris flow event in Jinyang County, China. Our analysis yields several key insights: (1) Spatial patterns: We identify approximately 3.2 × 10<sup>4</sup> high-risk gullies (each exceeding 200 m in length) predominantly located within a 30 km buffer zone along the middle and lower reaches of the Jinsha–Yalong River system. Notably, regions with infrequent extreme rainfall tend to exhibit larger debris flow volumes; (2) Dominant controls: The debris flow probability follows the hierarchy: surface erodibility > surface connectivity > stream power > extreme rainfall frequency > extreme rainfall intensity. This indicates that basin substrate characteristics exert stronger controls on debris flow development than climatic drivers in the study area; (3) Climate implications: Despite a decline in extreme rainfall events since 2014, the overarching trend of global warming persists, suggesting an eventual increase in extreme rainfall frequency. Over longer temporal scales, debris flow-prone zones are projected to migrate toward higher elevations. The resulting datasets—comprising fluvial power, surface connectivity, and debris flow probability maps—provide robust quantitative inputs for infrastructure siting, including hydropower hubs, transport corridors, and residential zones. Our process-informed "Geomorphic Dynamics—Spatial Patterns—Probabilistic Hazard" framework offers a transferable model for risk-tiered debris flow management in mountainous river systems worldwide.

## Data availability




The dataset we provide primarily includes the following information: (1) River Power and River Power Gradient Spatial Data for the Jinsha River Basin; (2) Surface Connectivity Spatial Data for the Jinsha River Basin; and (3) Debris Flow Occurrence Probability Maps for Small, Medium, and Large-Scale Events in the Jinsha River Basin. These datasets are integrated into a point grid format and provided in ".shp" format. The dataset is available as supplementary material for direct download, or may be requested by contacting the corresponding author.

# 500 Author contributions

Z. Gu was responsible for constructing the indicator system, conducting data analysis, producing visualizations, and drafting the manuscript; X. Yao provided field investigation data related to the "8.21" disaster event in Jinyang County; X. Zhu contributed soil data.

## **Competing interests**

The authors declare that they have no conflict of interest.

## Acknowledgements

We express our sincere gratitude to the anonymous reviewers for their meticulous suggestions. We also express our gratitude to Professor Changxing Shi from the Institute of Geographic Sciences and Natural Resources Research, Chinese Academy of Sciences, for his invaluable input on optimizing the key parameters in this study.

#### 510 Financial support

This work was supported by the National Natural Science Foundation of China (Grant NO. 42107218), the Project Gorges Corporation (Grant NO. YMJ(BHT)/(21)036), the Research Fund of the Institute of Geomechanics, CAGS (Grant No. 22511), China Geological Survey Project (Grant NO. DD20230433) and the Youth Innovation Promotion Association of the Chinese Academy of Sciences (Grant NO. 2023327).

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
