# Peer review of "Debris Flow Susceptibility in the Jinsha River Basin, China: A Bayesian Assessment Framework Based on Geomorphodynamic Parameters"

_EGUsphere, 2024_

## Author Comment (AC1)

**Manuscript Modifications: Point-by-point Responses**

**Dear Reviewers and Editors,**

Thank you very much for allowing us to revise our manuscript further. We would like to express our appreciation to you for your valuable comments and suggestions regarding our manuscript. We have made revisions following your comments and suggestions, and the revised contents are marked using the "Track Changes" function of Microsoft. You can view all changes using the "Display for Review" function of Microsoft Word. The line number corresponds to the revised manuscript without changes marked. We have tried our best to correct all grammatical mistakes and statement errors in the manuscript. Please see our point-by-point responses to the Editors' and Reviewers' comments below.

**Reviewer**

The authors present a methodology for assessing debris flow susceptibility over a large scale (study area size ~500000 km² ). The new method aims to focus heavily on the source-sink dynamics of debris flows rather than a more classical approach of surface characteristics. In my opinion, this and especially the inclusion of stream-power is innovative. Generally, the manuscript is well-written and concise, therefore I recommend the research to be published. The manuscript does require better descriptions on multiple aspects of the methodology though for which I recommend revisions. Below I first list my most pressing issues which require clarifications. This is followed specific comments and miscellaneous points. I'm happy to engage with the authors when they have answers or follow-up questions to my points.

**Comment 1:** Major points—For understanding of the reader, I think the methods benefit from a table where input factors to the Bayesian model are listed. This table should include description, resolution (if applicable), reference to details of the factor in the manuscript and preferably a range of values.

**Response 1:** Based on your suggestion, we have included an explanation of the model in the "Modeling Approach" section. The actual estimation process involves classification operations using a Gaussian distribution or other functions, where 0, 1, 2, and 3 represent different scale types. During the operation, we only need to set the number of categories, model method, and number of factors. We have provided the

original program and its documentation in the appendix to facilitate replication of the experiment by readers, addressing any potential gaps in the understanding of the methods presented in the paper.

**Comment 2:** The authors choose to focus on indicators of the source to sink characteristics of debris flows and mention that ignoring these characteristics results in low accuracy (without mentioning a source, that should be addressed as well). From the abstract: "Due to its vast area and the complex mechanisms underlying debris flow formation, using slope-based indicators alone to assess susceptibility, without considering the "source-sink" process of debris flow formation, results in low accuracy in susceptibility evaluations."

**Response 2:** During our research process, we reviewed numerous papers on the use of machine learning for debris flow hazard assessment. We identified a key issue: the indicators used in these studies are relatively basic and primarily describe fundamental geomorphological characteristics, such as slope and lithology. There is a lack of an indicator system that reflects the characteristics of physical processes, and the machine learning models are decoupled from the underlying physical principles. When we refer to "low assessment accuracy," we are not questioning the accuracy of the results but rather proposing an update to the assessment logic. After precipitation, debris material flows along the slope into the valley bottom, forming a material flow in the channel. This process involves some hidden physical principles: First, the larger the catchment area at a specific point in the channel, the larger the mass scale of the material flow passing through that point; second, for the material within the watershed to be transported into the riverbed as a source for debris flow, the surface material must be easily eroded, and the terrain must meet connectivity characteristics (for instance, flat areas on the slope can hinder the transport of debris from upstream to downstream, representing low terrain connectivity); third, a large channel gradient is not necessarily an indicator of frequent debris flows. The gradient must form a proper combination with the catchment area, precipitation, runoff, and upstream material supply to generate debris flow. In other words, to improve assessment accuracy, machine learning models must be designed based on the physical principles of "watershed erosion-transport-deposition," which incorporates physical model constraints into machine learning. In the revised manuscript, we have refined the wording, especially the explanation of the causes, to help readers understand (the modified text can be found in lines 4-7 of the

"Abstract").

**Comment 3:** I think the method in the manuscript, which neglects other factors of importance, could be partially responsible for their own relatively low accuracy. Factors such as vegetation, lithology and soil transmissivity (also mentioned by the authors for classical approaches, line 68) are what come to mind. I think deliberately neglecting these factors bends the aim of the manuscript from an overall debris flow hazard indicator to introducing a specific source-sink process-based method. This is still innovative and interesting, but I think the authors should mention their choices in this regard more explicitly at the end of the introduction and in the methods.

**Response 3:** Our fundamental approach is to constrain the design of the machine learning scheme using the basic principles of watershed erosion and transport. During the research framework design process, we did not overlook the role of vegetation. The reason for not deliberately incorporating vegetation data is that current DEM data products are generally based on InSAR satellite observation technology, which does not filter out the elevation affected by vegetation. The calculation process of the geomorphological connectivity index (IC value) is based on this type of DEM, and thus the resulting IC values naturally include spatial variations in surface connectivity caused by vegetation. We also did not consider using geological maps to describe lithology, as descriptions based on geological maps are typically qualitative, which is not conducive to a quantitative assessment process. In fact, regardless of lithology, loose surface soils and weathered layers are the key contributors to debris flow formation. Therefore, we introduced the erodibility factor, K, from the Universal Soil Loss Equation. This indicator reflects the degree to which the surface is prone to erosion and is only related to the properties of the soil or weathered layer itself. It is a quantitative metric with clear physical meaning, which facilitates a more rigorous quantitative assessment. To aid in reader understanding, we have added relevant explanations (the modified text can be found in lines 8-10 of "3.1 Data and Preprocessing").

**Comment 4:** The resolution of the analysis is unclear to me. The manuscript regularly mentions a minimum valley length of 200 m. With a DEM resolution of approximately 30 m these would be 6-7 grid cells. Can you reliably estimate your input and apply all your functions, which often require upstream and downstream values, for such small river reaches? After reading the 'data availability' section I wonder why the data is

at 90 m and not at the DEM resolution? This is the first time I read the analysis is not on the DEM resolution. This and possible resampling should be mentioned in the methods. The issue with the valley length applies even stronger if the analysis was performed at 90 m. If I am missing something please let me know, otherwise I think the authors should be clearer on this issue in their methods.

**Response 4:** The base data we used is the 1" DEM (with a resolution of approximately 30m). However, when calculating surface connectivity, a 3x3 grid window is required at a minimum. The resulting connectivity raster data has a resolution of 3" (approximately 90m). The catchment area calculations and river power-related computations are also based on the 30m resolution. However, the channel gradient involved in the process requires linear function fitting of several raster data points ($\geq 2$) to obtain the fitting coefficients. This is an upscaling process. The final result obtained is a point-grid representation of the river channel, where the distance between points is approximately 90m based on the resolution. For display convenience, we performed interpolation on these points, and the resolution can be adjusted according to the display requirements. Additionally, regarding the "200m" (6"), due to the large watershed area and the numerous valleys, we conducted a line connection process on the high-energy point-grid to convert them into line segments for counting the number of "high-energy valleys." We only need to count the line segments; however, if the distance between two points is less than 200m, and there is only one point within that distance, a line segment cannot be formed, and it will not be included in the statistics. (The modified text can be found in lines 17-18 of "4.1 Mapping of High-Energy Valleys and Erosion Dynamics").

**Comment 5:** Specific comments—Line 84/85. These numbers are suspicious. A length of 2316 km and an average gradient of 1.45% (~0.8 º) yields a vertical distance of ~ 33 km (length * tan(slope)). This is not very realistic, am I missing something?

**Response 5:** We greatly appreciate your thorough review. In the revised manuscript, we have corrected the data and symbols.

**Comment 6:** Line 105. For the ECMWF ERA5, a description of resolution, duration and related uncertainty are required as well as a reference.

**Response 5:** The suggested additions have been incorporated. (The modified text can be found in lines 106-108.)

**Comment 7:** Line 112. What threshold is used to define flat?

**Response 7:** When conducting hydrological analysis using the D8 algorithm, we need to perform depression filling. The areas that can be filled have a post-filling elevation change of zero. These areas are often reservoirs or ponds, where both debris flow and runoff experience energy dissipation as they pass through, making them irrelevant for analysis. During the slope calculation process, linear fitting results in null values for these locations. These areas are easy to identify.

**Comment 8:** Section 3.2. Why did the authors specifically choose a Naïve Bayesian model? And not for instance logistic regression or a random forest model? This choice should be clarified.

**Response 8:** The primary reason for selecting the Bayesian model is its result interpretability and the need for ease of expression and understanding. The Bayesian model provides a comprehensible form of results, specifically prior probabilities. This is important because we do not want to obtain probabilities only after a region has experienced multiple disaster events. Instead, we estimate the probability for a larger area based on the natural attributes of locations where disasters have occurred in the past. This feature is not available in other types of models. The reason we chose the Naive Bayes model is that the research scenario satisfies the strong assumption of "conditional independence between features," and the model itself has adaptability to small samples. It performs relatively well in scenarios with high feature dimensions, sparse data, or limited resources.

**Comment 9:** Line 136 Li

**Response 9:** We greatly appreciate your thorough review. In the revised manuscript, we have corrected the symbols.

**Comment 10:** Line 148. Higher than 0

**Response 10:** We have made improvements in the revised manuscript.

**Comment 11:** Line 150-160. Text doesn't read well. Could benefit from some critical rewriting.

**Response 11:** We have rewritten this section based on your suggestion. (The modified content can be found in lines 9-18 of "3.3.1 Stream power and its gradient.")

**Comment 12:** Line 159. $10^4$ I hope. Is this a 'guestimate' or did you calibrate? Be specific of how this was chosen. Also give the values of the a and b fitting parameters.

**Response 12:** This value is a reference value that we provided, based on a rough estimate considering typical debris flow density. Additionally, the determination of the

values for aaa and bbb was obtained through fitting observed data from several hydrological stations in the Jinsha River Basin, as shown in the corresponding figure in Panel 5d.

**Comment 13:** Line 220. Looks better when the formula is fitted on one line.

**Response 5:** The modifications have been made based on your suggestion.

**Comment 14:** Line 231. Correct me if I'm wrong, but I don't understand your statements. In Radoane et al. (2003) four functions (linear, exponential, log and power) are calibrated and tested for which can best describe the longitudinal profile of various rivers. They don't mention "progressions through stages of function curves". Clarify what you're trying to say here.

**Response 5:** The morphological changes in the longitudinal profile of a river are a key aspect of watershed geomorphological evolution. In the early stages of erosion, the longitudinal profile of the valley typically resembles a straight line. As development continues, it gradually becomes more curved, and ultimately, the mountains are eroded into a peneplain. At various stages of this process, the valley's longitudinal profile can be optimally fitted using four functions: linear, exponential, logarithmic, and power, in the following order: linear → exponential → logarithmic → power. (The modified content can be found in lines 4-9 of "4.1 Mapping of High-Energy Valleys and Erosion Dynamics.").

**Comment 15:** Line 242. Having done the GIS analysis, don't you have an exact number for how many valleys?

**Response 15:** We identify high-energy valleys by detecting outliers in stream power or stream power gradients. The revised manuscript includes supplementary figures and explains the method used to determine the threshold.

[Figure]

Figure 6: Characteristic statistics of High-Energy valley: (a)Variation in the Number of High-Energy Reaches with Channel Buffer Distance; (b) Debris flow investigation points in various stream power gradient intervals. Note: High-energy valleys are defined here as those with a stream power gradient greater than $1.3\times10^{-4}$W/m² and the threshold is defined according to the inflection point of the trend change of the fitted curve. This chart displays the count of high-energy valleys within a 200m buffer along the Jinsha River and Yalong River, across a range of buffer widths, specifically including those with a stream power gradient exceeding $1.3\times10^{-4}$ W/m².

**Comment 16:** Figure 5c. Add stream power units to the second axis on the right.

**Response 16:** We have made the necessary supplements as required.

**Comment 17:** Figure 5f caption should be: 'Photo by one of the authors' As there are multiple authors of the manuscript.

**Response 17:** We have made the necessary supplements as required.

**Comment 18:** Figure 6. Why this threshold for high-energy valleys?

**Response 18:** In the revised version, we added illustrations. Based on the illustrations, we set the position of the inflection point where the curve trend changes as the threshold for the division of high-energy river valleys.

**Comment 19:** Line 252. Any reason for this thresholding?

**Response 19:** As mentioned above.

**Comment 20:** Line 253. Does clay correlate with erodibility?

**Response 20:** They are related. In Formula 14, the calculation of the K value makes use of the clay content.

**Comment 21:** Figure 7 misses a legend on the right.

**Response 21:** This was our oversight. The image method overstepped the text boundaries. In the revised version, we re-adjusted the size of the image.

**Comment 22:** Line 269. Why is that suggested (This spatial shift suggests that the

pattern of extreme precipitation events is not stable over time.)?

**Response 22:** Due to spatial displacement of outliers in extreme precipitation events at different temporal observation scales (daily, monthly, yearly), extreme precipitation events tend to exhibit higher randomness at the daily scale. This randomness is often influenced by local climatic factors such as topography, temperature, and humidity. In contrast, the likelihood of extreme precipitation events at the yearly scale is more stable, primarily determined by fundamental factors such as the Earth's orbit. If the anomalous locations of extreme precipitation event characteristics do not change when observed across daily to yearly scales, it indicates minimal interference from local factors and high randomness, thus reflecting temporal and spatial stability, which is an external manifestation of the mechanisms behind extreme precipitation occurrences. (The modified text can be found in lines 5-8 of "4.3 Variations in Extreme Precipitation Events and Implications for Debris Flow Risk.")

**Comment 23:** Line 274/275. Are any of these statements statistically significant given the timeframe of 10 years?

**Response 23:** Since the correlation of the fitting function is significantly greater than the correlation threshold, it can be considered that this trend is significant.

**Comment 24:** Figure 10. The spatial pattern of the occurrence probability appears related to the extreme precipitation as one would expect. But this means the interpolation pattern is clearly visible. This makes me wonder if your rainfall data, SPI, really gives full spatial cover or whether it is an interpolation from specific sites.

**Response 24:** The influence of extreme precipitation is significant, but in the indicator system we constructed, its significance is slightly lower than that of other factors (Figure 11b). According to our analysis results, the geomorphological characteristics of the underlying surface play a more important role in the susceptibility to debris flow, with their importance slightly higher than that of extreme precipitation.

**omment 25:** Line 281: Move the accuracy to section 4.5. This is where you present the model verification.

**Response 25:** We have made adjustments based on your suggestion.

**Comment 26:** Figure 11. How is the relative importance calculated?

**Response 26:** In machine learning, if a change in the value of a particular factor leads to a more significant change in the dependent variable, then the relative importance of that factor is higher. This can be understood through a simpler example. For instance,

in multiple linear regression, each independent variable in the results corresponds to a significance level p-value. The smaller the p-value, the more significant the factor, and thus, the importance of the factors can be ranked based on the significance of the p-value.

**Comment 27:** Figure 12/13 (and in the conclusions). I appreciate the authors highlighting the impact a disaster like a debris flow can have. It stresses the importance of this type of research. I do wonder though how relevant 1 data point is when presenting a province-scale model and I would reduce the prominence of this description. I think it's more interesting is to have an overview of all the debris flows the authors collected and their occurrence probability. This will also help in interpreting figure 11 better.

**Response 27:** Thank you for your guidance. This typical disaster event, reported by the media, provides another crucial piece of evidence supporting our findings. It should be noted that such a catastrophic event, which causes significant loss of life and has severe social repercussions, is not something that occurs frequently within a region. Since the results of our study, we have only identified this single major disaster event. While considering the number of events, one instance might seem a bit few, we should also take into account the spatial performance of our results. The research precisely pinpointed the location of the disaster, which aligns closely with the actual location, an outcome that exceeded our initial expectations. The accuracy of this spatial localization is based on the thorough integration of physical process indicators during the model construction, which gives us confidence in its reliability.

**Comment 28:** Section 4.5 Is this the same dataset used to train the model, then the 70/30 split and compute accuracy? If so, the dataset should be mentioned and described in the methodology, not here.

**Response 28:** The data is the same; we have simply divided it randomly into two parts. (The modified text can be found in lines 1-2 of "4.5 Verification of Disaster Probability Maps with Actual Cases.")

**Comment 29:** Figure 13 b/f: The red text in the figure is unreadable.

**Response 29:** In the revised manuscript, we have replaced the figure.

**Comment 30:** Line 305. Was the actual event also a medium-scale debris flow according to your classification?

**Response 30:** Yes, however, due to the concentration of construction workers, the scale

of casualties resulted in a major disaster.

**Comment 31:** Line 323-326. I don't follow this reasoning. How is the observation timescale relevant to the distribution in precipitation event intensity. Isn't the goal of a statistical analysis, precisely not to have this effect? Am I misunderstanding something?

**Response 31:** In observing climate change, the time scales we choose for observation are associated with different levels of driving factors. Climate change at the yearly scale is often determined by ocean-atmosphere coupling oscillations, solar radiation variations, and other factors, while at the monthly scale, it is primarily influenced by the seasonal periodic changes in Earth's orbit. At the daily scale, climate change is affected by local factors such as solar radiation flux, valley wind circulation, and land-sea breeze oscillations. Extreme weather events are often the result of the interaction of multiple dynamic factors. Therefore, the driving forces reflected by extreme precipitation events observed under different temporal scales also vary. (The explanation has been added in lines 2-6 of "5.1 Impact of Temporal Observation Scale Changes on the Assessment" in the revised manuscript.)

**Comment 32:** Line 351. The higher CC-scaling for high-altitude areas. Does that also hold in your specific study area, where you state that it's the higher elevated areas that are the driest?

**Response 32:** Firstly, we acknowledge the scientific conclusions of previous studies, as similar findings have been widely reported and hold general significance. However, this does not mean that the watershed strictly follows this rule during the study period. In our research, at least since 2014, the frequency of extreme precipitation events has shown a decreasing trend. (This insight has been added in the revised manuscript.)

**Comment 33:** Line 365. Minimal interannual variability in the basin. How do you reconcile this with your own timeseries, Fig 9, showing large interannual variation at least in extreme precipitation events.

**Response 33:** "Minimal interannual variability in the basin" refers to precipitation in a broad sense, whereas what we emphasize in our study is the case of extreme precipitation events. The results presented in the paper are based on daily precipitation data for extreme precipitation events. In the revised manuscript, we have added the corresponding text to prevent any potential misunderstanding by the readers. (The modified text can be found in lines 283-284 of "5.2 Changes in Debris Flow Susceptibility Influenced by Climate Change.")

**Comment 34:** Line 370. Source?

**Response 34:** We have cited the literature as per your suggestion.

**Comment 35:** Line 376. Source?

**Response 35:** We have cited the literature according to your suggestion.

**Comment 36:** Line 385-390. Sources?

**Response 36:** We have cited the literature as per your suggestion.

**Comment 37:** Line 450. The distribution of the valleys? Like spatially?

**Response 37:** This sentence refers to Figure 6a. Due to the fragmented nature of the knowledge, we have removed this sentence in the revised manuscript.

**Comment 38:** Author contributions. I'm surprised to read about a campaign and measurements. Isn't the work for the large part a GIS + data gathering exercise?

**Response 38:** We have provided a more detailed supplement regarding the authors' contributions.

**Comment 39:** Technical corrections—Overall little spelling errors and absent/double spaces. Give it a thorough review on this.

**Response 39:** We have rechecked the text according to your suggestion.

**Comment 40:** With figures. Written text such as labels/axis titles is often uncomfortable to read due to small font.

**Response 40:** In the revised manuscript, we have provided clearer figures.

---

## Author Comment (AC4)

**Manuscript Modifications: Point-by-point Responses**

**Dear Reviewers and Editors,**

Thank you very much for allowing us to revise our manuscript further. We would like to express our appreciation to you for your valuable comments and suggestions regarding our manuscript. We have made revisions following your comments and suggestions, and the revised contents are marked using the "Track Changes" function of Microsoft. You can view all changes using the "Display for Review" function of Microsoft Word. The line number corresponds to the revised manuscript without changes marked. We have tried our best to correct all grammatical mistakes and statement errors in the manuscript. Please see our point-by-point responses to the Editors' and Reviewers' comments below.

**Reviewer #2**

The study tackles the relevant problem of regional debris-flow susceptibility. I think the authors chose adequate methods (Bayesian statistical methods) and an interesting combination of features representing hydro-geomorphological factors for debris-flow triggering. However, I found the methods to be intransparent in important acspects of the study, such that I cannot assess the reproducibility or plausibility of the results. I list my major concerns below.

**Comment 1:** What debris flow data was used for training, namely to estimate the occurrence probability in P(Ci) in Eq. 1? You mention that you use "debris flow survey sites" (L123) but there is no reference or description of how you obtain the data. Also Fig. 2 on the study implementation doesn't mention any use of observational debris flow data to train the model. In L115, I see one citation that may refer to such data (Yu and Tang, 2016), but the full reference is missing. Fig. 5 indicates, that debris -flow fans were identified, but how exactly and how do you differentiate debris-flow fans from alluvial fans?

**Response 1:** In the revised manuscript, we have provided a more detailed explanation of this section and included the corresponding MATLAB code. The survey data were obtained from the Ministry of Natural Resources of the People's Republic of China and are publicly accessible via the Resource and Environmental Science Data Platform of

the Chinese Academy of Sciences (https://www.resdc.cn/). Regarding references, we have added relevant literature to support and contextualize our methods. Our research is designed to identify a greater number of potential risk points by training on a limited set of fundamental survey locations, aiming to approximate a full-scale 1:1 mapping of debris flow susceptibility. This represents the practical significance of our work. Based on topographic maps and remote sensing imagery (see Fig. 1-2), we do not rely on all survey points as training data. Instead, we optimize data quality by selectively screening points using a range of attribute indicators, thereby maintaining the model error within a reasonable margin. This methodological refinement is now emphasized in the revised text. Although the scale of our survey map may not be exhaustive, the dataset remains objective and robust. In Fig. 5, the presence of debris flow fans serves as a geomorphic indicator to validate the threshold used for defining high-energy valleys. Specifically, the application of an average stream power gradient threshold of $1\times10^{-4}$ W/m² allowed us to effectively distinguish high-energy valleys, and the observed debris flow fans (Fig. 5f-1 and 5f-2) empirically confirm the suitability of this classification criterion. We have added corresponding annotations in the revised manuscript (see lines 13–14 of Section 3.1 "Data and Preprocessing").

[Figure]

Published Disaster Map

Fig. 1 Distribution of debris flow disaster sites

[Figure]

Fig. 2 Vector data

**Comment 2:** Any information on model training and testing is missing, except for the showcasing the model for one event

**Response 2:** Regarding the modeling process, it shares similarities with neural network approaches in that it operates as a "black-box" model. The core mechanism involves constructing probability distribution functions to generate probabilistic predictions. In the revised manuscript, we have elaborated further on the underlying principles of the model to enhance transparency and interpretability. These additions can be found in the revised text, specifically in lines 9–34 of Section 3.2 "Modeling Approach".

**Comment 3:** There is no uncertainty assessment or discussion of model limitation

**Response 3:** In the revised manuscript, we have added a discussion on the limitations of our modeling approach. These limitations mainly lie in two areas: First, the model assumes strict independence among input variables. This constraint implies that any inherent correlation between variables—especially those sharing a common physical mechanism—must be intentionally avoided during the construction of the indicator system. As a result, the selection of indicators is inherently restricted in both type and number, limiting the comprehensiveness of the parameter set. Second, while the model achieves reasonably accurate spatial predictions, it does not incorporate temporal information. It is important to note, however, that the parameters used in this study are derived from geomorphic processes and are intended to capture key dynamic elements of landscape evolution. These parameters possess clear physical meanings, and thus the identified high-probability zones represent locations of confirmed hazard potential,

despite the absence of precise timing. We have incorporated these points into both the "Modeling Approach" and "Conclusion" sections to more fully acknowledge the methodological limitations.

**Comment 4:** The data availability statement states that datasets are being made available, but there is no link. Anyway, more important would be the data to reproduce the results and this would include the debris flow observations

**Response 4:** We have uploaded the dataset and explicitly documented the source of the survey data.

**Comment 5:** The conclusions are largely a copy of the abstract. Both should be rewritten such that they are complementary (e.g., more focus on research question and methods in abstract and more focus on conclusions, implications, outlook in conclusions)

**Response 5:** We have revised and rewritten both the abstract and the conclusion sections.

**Comment 6:** Specific comments, ~L70-77 : I cannot follow the critique on previously used indicators for DF susceptibility. It may be that the risk is highest in the valley bottom, the source area characteristics govern susceptibility. Can you specify what current methods exactly are missing and what you do differently? Contradictory to your argument on the importance of valley bottom characteristics, I would assume that the factors you report in L77 (stream power, surface erosion, etc) characterize source are rather than sink area.

**Response 6:** Our original expression in the manuscript may have lacked clarity. In response to your suggestion, we have revised this section to provide a more precise explanation. Our intention was to emphasize the need for a more targeted indicator system—one that specifically focuses on the linear river channels at the bottom of gullies. This system is designed to directly capture the physical and energetic characteristics of valley floors. In contrast, the broader hillslope source areas generally exhibit more subdued material transport processes. By accurately characterizing the matter–energy dynamics at the gully base, we aim to enhance the predictive power of the model, particularly in terms of spatial localization accuracy.

**Comment 7:** L105: if you use ERA5, higher resolutions that daily are available to my knowledge. Could you justify why you don't use these? Sub-daily rainfall is commonly much more useful than daily for debris-flow triggering

**Response 7:** While the ERA5 dataset provides precipitation records at hourly

resolution, our research is primarily concerned with improving the spatial accuracy of debris flow susceptibility predictions, rather than modeling the triggering process of individual debris flows at the gully scale. Given the relatively broad spatiotemporal scale of our analysis, daily-scale precipitation data are sufficient to address the scientific questions posed in this study.

---

## Referee Report (RR1)

The Authors did a good job in addressing my review comments one by one and in adapting the manuscript. Of my original comments, there are two remaining where I require a clarification and an adjustment of the manuscript. These are given below. Other issues are solved from my point of view.

Comment 3: I think the method in the manuscript, which neglects other factors of importance, could be partially responsible for their own relatively low accuracy. Factors such as vegetation, lithology and soil transmissivity (also mentioned by the authors for classical approaches, line 68) are what come to mind. I think deliberately neglecting these factors bends the aim of the manuscript from an overall debris flow hazard indicator to introducing a specific source-sink process-based method. This is still innovative and interesting, but I think the authors should mention their choices in this regard more explicitly at the end of the introduction and in the methods.

Response 3: Our fundamental approach is to constrain the design of the machine learning scheme using the basic principles of watershed erosion and transport. During the research framework design process, we did not overlook the role of vegetation. The reason for not deliberately incorporating vegetation data is that current DEM data products are generally based on InSAR satellite observation technology, which does not filter out the elevation affected by vegetation. The calculation process of the geomorphological connectivity index (IC value) is based on this type of DEM, and thus the resulting IC values naturally include spatial variations in surface connectivity caused by vegetation. We also did not consider using geological maps to describe lithology, as descriptions based on geological maps are typically qualitative, which is not conducive to a quantitative assessment process. In fact, regardless of lithology, loose surface soils and weathered layers are the key contributors to debris flow formation. Therefore, we introduced the erodibility factor, K, from the Universal Soil Loss Equation. This indicator reflects the degree to which the surface is prone to erosion and is only related to the properties of the soil or weathered layer itself. It is a quantitative metric with clear physical meaning, which facilitates a more rigorous quantitative assessment. To aid in reader understanding, we have added relevant explanations (the modified text can be found in lines 8-10 of "3.1 Data and Preprocessing").

Subsequent Response reviewer 3: Correct me if I'm misinterpreting here, but it reads as if you treat canopy height as an addition to the DEM. This is not how I think vegetation should be included. Vegetation has a complex interaction with soil hydrology and geotechnics. It is not 'additional elevation'. My advice would be to somehow reflect vegetation presence in your model as independent variable or incorporate it in the erodibility/connectivity. Another option would be to ignore it of course, as it might not be a focus of the study. I think you should also mention that the DEM you use is in fact a Digital Surface Model.

Comment 26: Figure 11. How is the relative importance calculated?

Response 26: In machine learning, if a change in the value of a particular factor leads to a more significant change in the dependent variable, then the relative importance of that factor is higher. This can be understood through a simpler example. For instance, 9 in multiple linear regression, each independent variable in the results corresponds to a significance level p-value. The smaller the p-value, the more significant the factor, and thus, the importance of the factors can be ranked based on the significance of the p-value.

Response reviewer: Mention this method explicitly, or with a reference.

---

## Referee Report (RR2)

**General comments:**

In my opinion the paper provides innovative methodology to model debris flow susceptibility. Overall, the paper is good shape with concise language and descriptions. However, I found it sometimes very hard to read and understand the figures. There is a lot of information on the figures with small fonts, I suggest increasing the size of the fonts in the figures for better readability. Additionally, I wonder if some subplots could be removed in order to focus more specifically on key aspects of the figures.

Furthermore, I found the method and result section about extreme precipitation difficult to understand. Specifically, I did not understand how the computed severity relates to the Standardized Precipitation Index (SPI) of Table 1? And how is heavy precipitation (used in Fig 8) defined? Can you better explain these sections?

**Specific comments:**

- Table 1 is not referenced in the text.
- Sometimes spaces are missing in the text. On following lines, I found missing spaces. But please carefully check the entire manuscript for further missed or double spaces.
  - o Lines: 41, 103, 182, 185, 217, 229, 456
- Line 159: I suggest writing 1.6 \* 104 N/m3 as 16,000 N/m3
- Figure 5 misses the index f-1
- Figure 8 misses a description of subfigures b, e, h in the caption
- Line 299 state the year of the date (2024)
- The sentence of lines 333-335 misses a references
- The sentences about climate of lines 354 358 also need references.

---

## Author Response (AR2)

**Manuscript Modifications: Point-by-point Responses**

**Dear Reviewers and Editors,**

Thank you very much for allowing us to revise our manuscript further. We would like to express our appreciation to you for your valuable comments and suggestions regarding our manuscript. We have made revisions following your comments and suggestions, and the revised contents are marked using the "Track Changes" function of Microsoft. You can view all changes using the "Display for Review" function of Microsoft Word. The line number corresponds to the revised manuscript without changes marked. We have tried our best to correct all grammatical mistakes and statement errors in the manuscript. Please see our point-by-point responses to the Editors' and Reviewers' comments below.

**Reviewer #1**

**Comment 1:** The Authors did a good job in addressing my review comments one by one and in adapting the manuscript. Of my original comments, there are two remaining where I require a clarification and an adjustment of the manuscript. These are given below. Other issues are solved from my point of view.

**Response 1:** Thank you for your positive feedback and the constructive suggestions for improvement. We have made the revisions in accordance with your recommendations and have added the appropriate references.

Comment 2: "Comment 3: I think the method in the manuscript, which neglects other factors of importance, could be partially responsible for their own relatively low accuracy. Factors such as vegetation, lithology and soil transmissivity (also mentioned by the authors for classical approaches, line 68) are what come to mind. I think deliberately neglecting these factors bends the aim of the manuscript from an overall debris flow hazard indicator to introducing a specific source-sink process-based method. This is still innovative and interesting, but I think the authors should mention their choices in this regard more explicitly at the end of the introduction and in the methods.

Response 3: Our fundamental approach is to constrain the design of the machine learning scheme using the basic principles of watershed erosion and transport. During

the research framework design process, we did not overlook the role of vegetation. The reason for not deliberately incorporating vegetation data is that current DEM data products are generally based on InSAR satellite observation technology, which does not filter out the elevation affected by vegetation. The calculation process of the geomorphological connectivity index (IC value) is based on this type of DEM, and thus the resulting IC values naturally include spatial variations in surface connectivity caused by vegetation. We also did not consider using geological maps to describe lithology, as descriptions based on geological maps are typically qualitative, which is not conducive to a quantitative assessment process. In fact, regardless of lithology, loose surface soils and weathered layers are the key contributors to debris flow formation. Therefore, we introduced the erodibility factor, K, from the Universal Soil Loss Equation. This indicator reflects the degree to which the surface is prone to erosion and is only related to the properties of the soil or weathered layer itself. It is a quantitative metric with clear physical meaning, which facilitates a more rigorous quantitative assessment. To aid in reader understanding, we have added relevant explanations (the modified text can be found in lines 8-10 of "3.1 Data and Preprocessing"). Subsequent Response reviewer 3: Correct me if I'm misinterpreting here, but it reads as if you treat canopy height as an addition to the DEM. This is not how I think vegetation should be included. Vegetation has a complex interaction with soil hydrology and geotechnics. Itis not 'additional elevation'. My advice would be to somehow reflect vegetation presence in your model as independent variable or incorporate it in the erodibility/connectivity. Another option would be to ignore it of course, as it might not be a focus of the study. I think you should also mention that the DEM you use is in fact a Digital Surface Model."

**Response 2:** The approach to fine-scale simulation typically aims to consider as many factors as possible to minimize errors. However, models built with this philosophy often face challenges in scaling to larger spatial assessments due to their computational intensity. We do not intentionally overlook any factors; rather, starting from general physical processes of surface dynamics, we aim to identify the most critical factors. This approach allows us to achieve high-quality results while greatly reducing data

processing demands, making it suitable for large-scale evaluations at the scale of thousands to tens of thousands of square kilometers.

The topographic data used in this study were not subjected to "vegetation elevation correction." Consequently, the surface connectivity derived from this data inherently includes the influence of vegetation, which we treat as a parameter for surface roughness. Furthermore, the water-soil coupling process is highly complex, with vegetation playing an integral role, often acting as an obstacle in the "source-sink" process. The dynamics between water and soil are more central to discussions on the driving environmental processes. In addition, the primary influence of vegetation on the formation of mountain floods and debris flows occurs on slopes, while our indicator specifically targets the valley floor, where the role of vegetation is relatively indirect. As such, vegetation was not singled out in the initial model. In the revised manuscript, we have supplemented the discussion on the role of vegetation while also emphasizing the focus of this study.

**Comment 3:**

Comment 26: Figure 11. How is the relative importance calculated?

Response 26: In machine learning, if a change in the value of a particular factor leads to a more significant change in the dependent variable, then the relative importance of that factor is higher. This can be understood through a simpler example. For instance, 9 in multiple linear regression, each independent variable in the results corresponds to a significance level p-value. The smaller the p-value, the more significant the factor, and thus, the importance of the factors can be ranked based on the significance of the p- value.

Response reviewer: Mention this method explicitly, or with a reference.

**Response 3:** The relative importance of a factor can be determined by calculating the difference in the log-likelihood ratios for different factors. The mutual information measure permits analysis with both continuous and categorical variables and has been widely adopted in the literature; we therefore select this metric (Blanquero et al., 2021). It quantifies the information about variable *X* contained in variable *Y*, defined formally

$$I(X,Y) = \iint P(x,y) \log \left(\frac{P(x,y)}{P(x)P(y)}\right) dxdy \tag{6}$$

In addition, we have provided clear references with explicit annotations in the revised manuscript (Blanquero et al., 2021).

**Reviewer #2**

**Comment 1:**

In my opinion the paper provides innovative methodology to model debris flow susceptibility. Overall, the paper is good shape with concise language and descriptions. However, I found it sometimes very hard to read and understand the figures. There is a lot of information on the figures with small fonts, I suggest increasing the size of the fonts in the figures for better readability. Additionally, I wonder if some subplots could be removed in order to focus more specifically on key aspects of the figures. Furthermore, I found the method and result section about extreme precipitation difficult to understand. Specifically, I did not understand how the computed severity relates to the Standardized Precipitation Index (SPI) of Table 1? And how is heavy precipitation (used in Fig 8) defined? Can you better explain these sections?

Response 1: The observed image degradation likely resulted from lossy compression during file format conversion; the original vector graphics have been restored in the manuscript. Additionally, regarding the severity of extreme precipitation, we have included a clear definition formula (18) in the Methods section, which represents the average SPI value during the duration of extreme precipitation events. Furthermore, Table 1 outlines the classification standards for precipitation intensity levels, including the categorization for extreme drought conditions. In addition, we have redrawn some of the illustrations (Fig. 8).

**Comment 2:**

**Specific comments:**

- Table 1 is not referenced in the text.
- Sometimes spaces are missing in the text. On following lines, I found missing spaces. But please

carefully check the entire manuscript for further missed or double spaces. Lines:

41, 103, 182, 185, 217, 229, 456

- Line 159: I suggest writing 1.6 \* 104 N/m3 as 16,000 N/m3
- Figure 5 misses the index f-1
- Figure 8 misses a description of subfigures b, e, h in the caption
- Line 299 state the year of the date (2024)
- The sentence of lines 333-335 misses a references
- The sentences about climate of lines 354 358 also need references.

**Response 2:**

We greatly appreciate your meticulous attention to detail, which has significantly enhanced the quality of our work. In response to your suggestions, we have made the following revisions:

Added labels for Table 1.

Conducted a thorough review to eliminate duplicate spaces.

Corrected the notation of  $1.6 * 10^4 \text{ N/m}^3$  to  $16,000 \text{ N/m}^3$ .

Revised the figure legend for Figure 5f-1.

Updated the figure legend for Figure 8 to include descriptions of subfigures b, e, and h.

Incorporated references to relevant literature on news events.

Included additional references concerning climate change.

Blanquero, R., Carrizosa, E., Ramírez-Cobo, P., Sillero-Denamiel, M.R., 2021. Variable selection for Naïve Bayes classification. Computers & Operations Research, 135, 105456.

---

## Author Response (AR3)

**Manuscript Modifications: Point-by-point Responses**

**Dear Reviewers and Editors,**

Thank you very much for allowing us to revise our manuscript further. We would like to express our appreciation to you for your valuable comments and suggestions regarding our manuscript. We have made revisions following your comments and suggestions, and the revised contents are marked using the "Track Changes" function of Microsoft. You can view all changes using the "Display for Review" function of Microsoft Word. The line number corresponds to the revised manuscript without changes marked. We have tried our best to correct all grammatical mistakes and statement errors in the manuscript. Please see our point-by-point responses to the Editors' and Reviewers' comments below.

**Yves Bühler**

Public justification (visible to the public if the article is accepted and published):

Dear Authors

Thank you for revising your manuscript. I have two issues that have to be resolved before your paper can be accepted for publication:

**Comment 1:** The data necessary to reproduce your study has to be made available for interested researchers. "Interested readers can request access to the dataset free of charge from the corresponding author" is not enough. The data should be downloadable on an accessible portal.

**Response 1:** As an institutional data repository is currently unavailable, the processed datasets have been provided as supplementary files during the manuscript revision process. Should the journal be unable to host downloadable files, we will direct readers to obtain these datasets via direct request to the corresponding author. These represent the primary data access modalities supported at present.

Comment 2: The author's contributions of R. Jiang, J. Shu and F. Dai of proofreading the manuscript do not justify a co-authorship following international standards. They should be removed from the list of authors

**Response 2:** Author contributions have been revised in strict adherence to international authorship criteria you recommended, with removal of non-contributing individuals from the byline.

Additional revisions: The study area map was replaced to correct annotation inaccuracies, with projected coordinate metadata now explicitly documented in the figure legend.